# On-surface synthesis of planar dendrimers via divergent cross-coupling reaction

Deng-Yuan Li[1], Shi-Wen Li[1], Yu-Li Xie[1], Xin Hua [1], Yi-Tao Long [1], An Wang[1] & Pei-Nian Liu [1]

Dendrimers are homostructural and highly branched macromolecules with unique dendritic effects and extensive use in multidisciplinary fields. Although thousands of dendrimers have been synthesized in solution, the on-surface synthetic protocol for planar dendrimers has never been explored, limiting the elucidation of the mechanism of dendritic effects at the single-molecule level. Herein, we describe an on-surface synthetic approach to planar dendrimers, in which exogenous palladium is used as a catalyst to address the divergent cross-coupling of aryl bromides with isocyanides. This reaction enables one aryl bromide to react with two isocyanides in sequential steps to generate the divergently grown product composed of a core and two branches with high selectivity and reactivity. Then, a dendron with four branches and dendrimers with eight or twelve branches in the outermost shell are synthesized on Au(111). This work opens the door for the on-surface synthesis of various planar dendrimers and relevant macromolecular systems.

[1] Key Laboratory for Advanced Materials and Feringa Nobel Prize Scientist Joint Research Center, State Key Laboratory of Chemical Engineering, School of Chemistry and Molecular Engineering, East China University of Science & Technology, 130 Meilong Road, Shanghai 200237, China. Correspondence and requests for materials should be addressed to P.-N.L. (email: liupn@ecust.edu.cn)

Dendrimers are a unique class of nanosized macro-molecules featuring homostructural and highly branched architectures[1]. As shown in Fig. 1a, the structure of a dendritic macromolecule is composed of three main parts: (1) a central core moiety, (2) interior layers made of divergently growing branch units attached to the core, and (3) terminal functionalities distributed at the outermost space. Thus, repeated fine-tuning of architectural variations enables the optimization of dendrimers for specific applications by the introduction of different functional moieties.

Ever since the first cascade molecules, and nee dendrimers reported by Vögtle et al. in 1978[2], thousands of dendrimers have been developed in solution[3–5], which exhibit a variety of unique dendritic effects and lead to extensive use in multidisciplinary fields[6–12]. In general, the synthesis of a dendrimer starts from a functionalized core and proceeds by divergent growth via sequential reactions. The growing reactions should generate two or more new branches from a single reactive site to form the dendritic architecture, and the reaction selectivity and reactivity

**Fig. 1** On-surface synthesis of planar dendrimers via DCR. **a** The general structure of a dendrimer. **b** On-surface coupling reactions. **c** On-surface synthesis of planar dendron (**2**) with four branches and dendrimers (**3** and **4**) with eight and twelve branches (the products might be surface-bonded radicals or hydrogenated species)

must be very high to guarantee the complete conversion of multiple reactive sites to avoid defects.

Dendrimers synthesized in solution stretch in three-dimensions and the configurations of the numerous components are very complicated and chaotic. In contrast, planar dendrimers synthesized in situ on surfaces could afford unambiguous configurations in the two-dimensional confined space, which are beneficial to elucidate the mechanism of dendritic effects at the single-molecule level and produce highly ordered, self-assembled structures. In addition, the in situ synthesis of dendrimers on surfaces solves the problems of solubility and sublimation caused by their high molecular weight.

However, the synthesis of planar dendrimers on surfaces has not been achieved yet due to the lack of a practicable on-surface synthesis method. The main challenges include (1) development of an on-surface divergent cross-coupling reaction (DCR), which enables the generation of two or more new branches from one reactive site, (2) achievement of sufficiently high selectivity of the cross-coupling with complete inhibition of the homo-coupling, and (3) achievement of sufficiently high reactivity of the cross-coupling to ensure complete conversion of the multiple reactive sites.

After a pioneering study in 2007[13], on-surface homo-coupling reactions have been extensively developed in various versions (Fig. 1b, I)[14], including dehalogenations[15–20], decarboxylations[21], and dehydrogenations[22–26], among others[27–29]. In addition, on-surface cross-coupling reactions have also been achieved (Fig. 1b, II), mainly involving the reaction of aryl bromides with porphyrin bromides[30], alkynes[31–35], and alkenes[36], although the reaction selectivity is usually poor. So far, on-surface coupling reactions are all limited in their reaction pattern to the generation of only one new covalent bond, which can only connect one fragment (**A**) with another fragment (**A** or **B**) to generate **A**–**A** or **A**–**B** structures (Fig. 1b, I and II). An on-surface coupling reaction generating two new bonds to connect one **A** with two **B** groups to form **A**–**2B** has never been reported (Fig. 1b, III).

Isocyanides are a unique class of building blocks in organic synthesis. Their terminal carbon atom has one pair of electrons and an empty orbital, therefore, enabling reactions with both nucleophiles and electrophiles at the same carbon atom. The reactions of isocyanides in solution have been extensively explored[37], but they have never been used in on-surface coupling reactions to date.

Herein, we report a cross-coupling of aryl bromides with isocyanides on a Au(111) surface using exogenous palladium as a catalyst. Surprisingly, one aryl bromide can react with two isocyanides to generate a divergent cross-coupling product (**1**) with one core and two branches. Due to the high selectivity and reactivity of this reaction, a planar dendron (**2**) with four isocyanide-derived branches and dendrimers (**3** and **4**) with eight and twelve isocyanide-derived branches have been successfully synthesized on a surface (Fig. 1c). These products can undergo further self-assembly to form highly ordered structures.

## Results

**On-surface DCR on Au(111).** As the first step of our investigations, 5-(4-bromophenyl)-10,15,20-triphenylporphyrin (**Br-TPP**) and 4-isocyano-1,1′-biphenyl (**ICBP**) were selected as the precursors (Fig. 2a) for the on-surface reaction on Au(111). The experimental results showed that no cross-coupling occurred between the porphyrin derived aryl bromide and isocyanide on Au(111) after annealing to 403 K (see Supplementary Fig. 2). It is noteworthy that the homo-coupling of aryl bromide or isocyanide could also not be observed, and isocyanide desorbed at this

annealing temperature[38]. Palladium is one of the most efficient and versatile catalysts in solution, especially in cross-coupling reactions[39]. So, exogenous Pd was tested as the catalyst for the reaction of aryl bromide and isocyanide, with the hope of enhancing cross-coupling activity while inhibiting the homo-coupling of the two precursors.

A submonolayer of **Br-TPP** molecules (see Supplementary Fig. 3a) and an excess of **ICBP** molecules (see Supplementary Fig. 4a) were successively deposited onto a pristine Au(111) surface (see Supplementary Fig. 4b) maintained at approximately 250 K inside a commercial ultra-high vacuum (UHV) system (base pressure: $\sim 2 \times 10^{-10}$ mbar) equipped with a variable-temperature scanning tunneling microscope (STM; SPECS, Aarhus 150). Pd atoms were subsequently dosed onto the Au (111) surface (see Supplementary Fig. 4c). The results demonstrate that the alternative arrangement of **ICBP** and **Br-TPP** was mostly retained after the Pd deposition and Pd clusters scattered in the molecules. Such miscibility of the reactants is crucial for the high selectivity of the DCR. After annealing at 403 K for 1 h, the sample was cooled to 120 K for STM analysis. Surprisingly, no homo-coupling products were observed under these conditions, and the DCR demonstrates high selectivity and reactivity. These results demonstrate that exerting kinetic control via diluted reactant conditions inhibits homo-coupling products, thereby kinetically selecting the DCR product.

Figure 2b shows the close-packed product molecules generated from the reaction of **Br-TPP** and **ICBP**, arranged in a herringbone-like pattern. The resulting highly ordered self-assembly tends to align with [11$\bar{2}$] and the equivalent directions of the substrate. The high-resolution STM images (Fig. 2c–e) reveal that the backbone of the aligned structure can be attributed to the porphyrin moiety (circled by the blue line in Fig. 2c) derived from the **Br-TPP** precursor. One V-shaped branch (circled by the white line in Fig. 2c) is attached to the porphyrin moiety, originating from two **ICBP** precursor molecules. Interestingly, the products were observed in a staggered arrangement, with the V-shaped branches arrayed on both sides. The structural characteristics of the product suggest that the divergent cross-coupling of one **Br-TPP** molecule with two **ICBP** molecules occurred on Au(111) to generate **1** with one core and two branches. Through the comparison of experimental STM images with the simulated images based on density functional theory (DFT) calculations (the adsorption models and simulated image, see Supplementary Fig. 5), the structure of **1** is proposed as indicated in Fig. 2a.

It is noteworthy that **1** is a prochiral molecule. According to the direction of rotation of the two branches on the surface, we defined the anti-clockwise species as S-**1** and the clockwise species as R-**1** (Fig. 2a). Figure 2c, d shows the two pure phases of self-assembled S-**1** and R-**1**, respectively. The self-assembly pattern mixing S-**1** and R-**1** could be found occasionally (Fig. 2e), but no pure phase could form such a pattern, probably due to self-assembly mismatching.

To further demonstrate the selectivity of the DCR, we performed statistical analysis for the systematic exploration of the coverage of **Br-TPP**, annealing temperature and time, in which homo-coupling product was not counted due to the low abundance of <2% (Fig. 2f–h). As shown in Fig. 2f, the abundance of **1** slightly decreased when the coverage of **Br-TPP** increased from 0.2 ML to 0.6 ML. Once the coverage increased to 0.8 ML, the abundance of **1** dramatically decreased and most molecules were **TPP** species without coupling, accompanied by a very small amount of homo-coupling product **TPP-TPP** (see Supplementary Fig. 6). This result might be attributed to the high surface coverage that led to the compactly packed porphyrins and

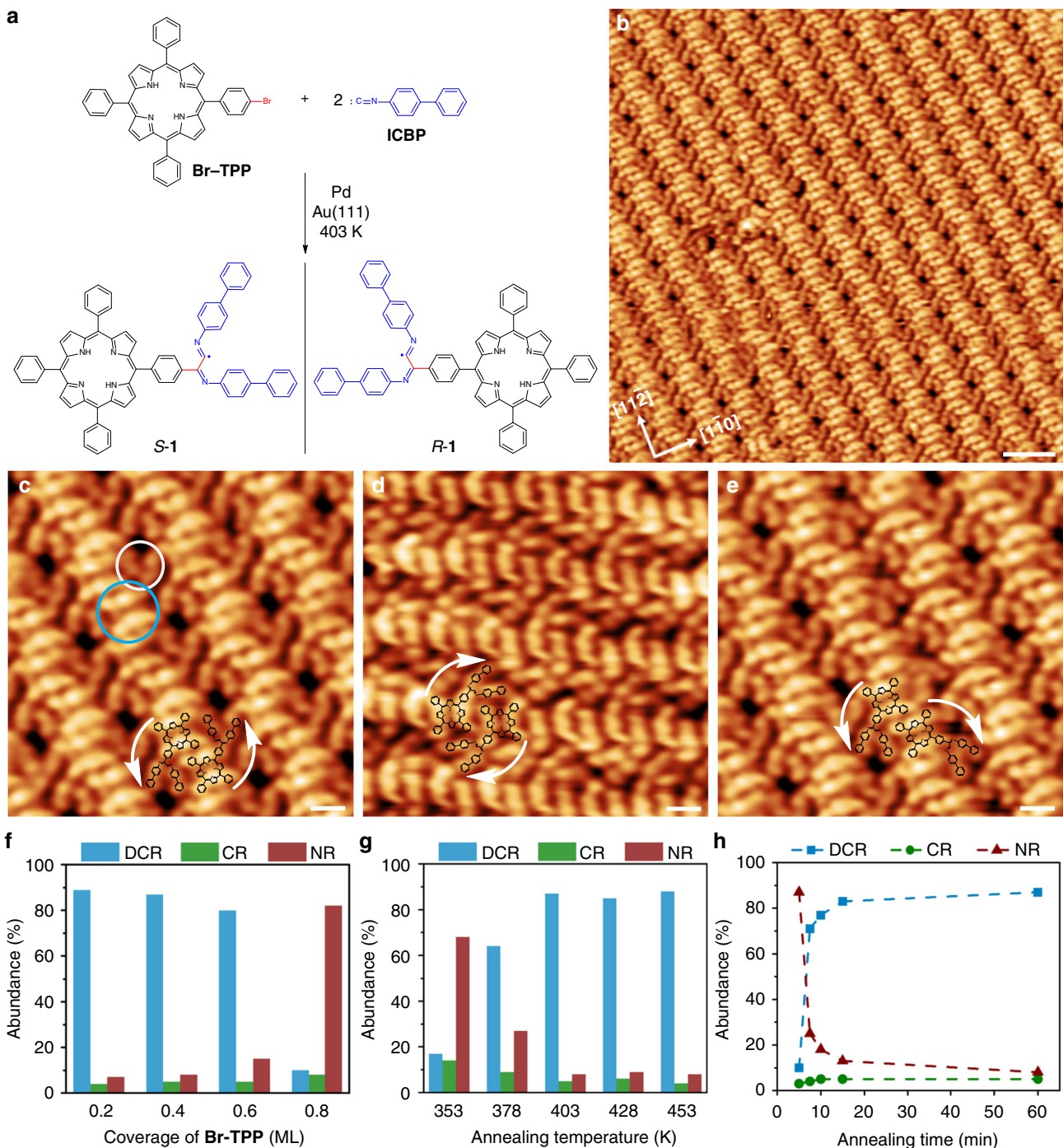

**Fig. 2** DCR of one **Br-TPP** with two **ICBP** on Au(111). **a** DCR used to generate *S*-**1** and *R*-**1**. **b** STM image of the product generated from DCR. **c** STM image of self-assembled molecules of *S*-**1**. **d** STM image of self-assembled molecules of *R*-**1**. **e** STM image of mixed molecules of *S*-**1** and *R*-**1**. **f–h** Statistical analysis for the DCR of **Br-TPP** with **ICBP** at different coverages of **Br-TPP** (**f**, reaction conditions: 403 K, 1 h), different annealing temperatures (**g**, reaction conditions: ~0.4 ML **Br-TPP**, 1 h), and different annealing time (**h**, reaction conditions: ~0.4 ML **Br-TPP**, 403 K). DCR: the reaction of one **Br-TPP** with two **ICBP**. CR: the reaction of one **Br-TPP** with one **ICBP**. NR: no coupling reaction. Scale bars: **b** 3 nm. **c–e** 1 nm. Tunneling parameters: **b**, **c**, **e** $I = -0.49$ nA, $U = -1.72$ V. **d** $I = -0.66$, $U = -1.78$ V

hampered both the cross- and homo-coupling reactions. Figure 2g shows that an annealing temperature of 378 K led to a slightly lower abundance of **1**, while lowering the annealing temperature to 353 K dramatically decreased the abundance of **1**. In contrast, enhancement of the annealing temperature demonstrated no obvious influence on the reaction. The results indicated that the reaction possesses good tolerance to the coverage of **Br-TPP** and annealing temperature. The investigation of annealing time at

403 K showed that the reaction was rather fast, and the abundance of the product was 77% after 10 min and increased to 82% after 15 min (Fig. 2h). The control experiment for Pd-catalyzed homo-coupling of **Br-TPP** without **ICBP** at 403 K for 15 min was carried out and approximate 10% abundance of homo-coupling product **TPP-TPP** was obtained in lower coverage of **Br-TPP** (~0.2 ML), similar with our former study performed in Omicron system[40], although the system error in

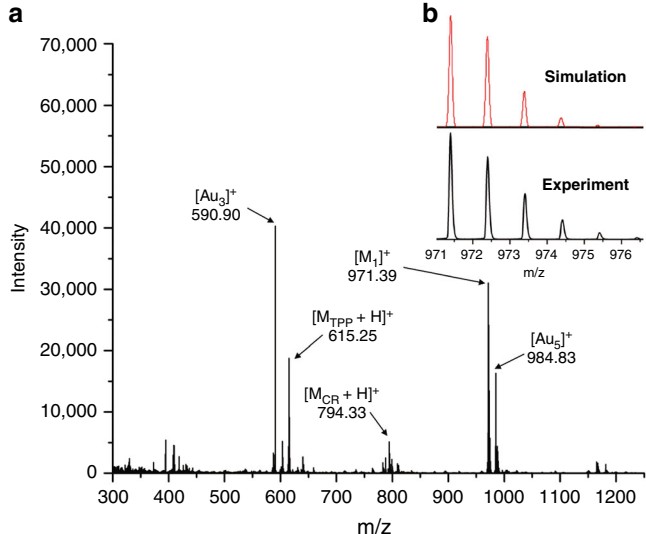

**Fig. 3** ToF-SIMS analysis of **1** on Au(111). **a** ToF-SIMS spectrum showing the molecular weight of positive ions (coverage of **Br-TPP**: ~0.5 ML). M**1**: molecular weight of divergent cross-coupling product **1**; M$_{CR}$: molecular weight of cross-coupling product of one **Br-TPP** with one **ICBP**; M$_{TPP}$: molecular weight of debrominated product of **Br-TPP**. **b** Experimental and simulated isotopic distribution of [M**1**]$^+$ ions of **1**

temperature measurement may exist between two setups. Moreover, the higher coverage of **Br-TPP** (~0.8 ML) led to lower abundance of **TPP-TPP** due to the surface coverage. The results suggested that the homo-coupling reaction of **Br-TPP** is sluggish, because the C–C bond formation, rather than the debromination step, is the rate-limiting step[40]. In contrast, the coupling step in DCR is fast, which guarantees the high selectivity of the reaction with the inhibition of the homo-coupling reactions.

To further elucidate the structure of **1** on Au(111), ex situ time-of-flight secondary ion mass spectrometry (ToF-SIMS) was performed to determine the molecular weight of the products. Figure 3a shows the mass spectrum of the positive ions obtained from a sample of the Pd-catalyzed cross-coupling on Au(111) with an m/z range from 300 to 1250. The distinct peak at m/z = 971.39 amu corresponds accurately to the molecular ion peak of **1** ($C_{70}H_{47}N_6^+$ [M**1**]$^+$, the calculated value is 971.39), and the experimental isotopic distribution of [M**1**]$^+$ agrees well with the theoretical isotopic distribution (Fig. 3b). According to the general principles of molecular ionization for ToF-SIMS experiments, hydrogenated product of **1** on Au(111) may lose an electron or gain a proton to generate positive ions[41], which would afford the corresponding peaks at m/z = 972.39 or 973.40, respectively. The peak observed at m/z = 971.39 suggests that **1** should consist of a radical bonded with the Au(111) surface rather than a hydrogenated product.

It is noteworthy that the lower abundance peaks at m/z = 794.33 and 615.25 amu correspond accurately to the quasi-molecular ion peaks (see Supplementary Fig. 7) for the partially branched product of the cross-coupling of one **Br-TPP** with one **ICBP** ($C_{57}H_{40}N_5^+$, [M$_{CR}$ + H]$^+$, calculated value is 794.33) and the debrominated product of **Br-TPP** ($C_{44}H_{31}N_4^+$, [M$_{TPP}$ + H]$^+$, calculated value is 615.25), respectively. In contrast, molecular ion peak of the homo-coupling product of **Br-TPP** ($C_{88}H_{59}N_8^+$ [M$_{TPP-TPP}$ + H]$^+$, the calculated value is 1227.49) was not observed (see Supplementary Fig. 8).

**Mechanism of on-surface DCR**. In order to unravel the reaction mechanism and elucidate the origin of the high selectivity of

DCR, we performed DFT calculations to determine the energy of the initial debromination and two subsequent addition steps using simplified bromobenzene and isocyanobenzene as model precursors (Fig. 4). Figure 4a shows the Pd adatom-promoted debromination step from bromobenzene (**IS1**) to a benzene radical intermediate (**Int1**) on the surface of Au(111). The reaction energy barrier was determined to be 0.70 eV, which is in good agreement with the experimental reaction temperature. In comparison, the reaction energy barrier for the debromination step without Pd on Au(111) is 1.02 eV[42], which indicates the key role played by the exogenous Pd catalyst to decrease the reaction barrier in the debromination step.

Figure 4b shows the first addition process of a benzene radical to isocyanobenzene in order to generate an imidoyl radical intermediate (**Int2**). The reaction energy barrier assisted by the Au(111) surface was found to be 0.24 eV, suggesting that this addition occurs spontaneously after the debromination step. For the Pd adatom-promoted addition of a benzene radical to isocyanobenzene, a higher reaction barrier energy of 0.46 eV (see Supplementary Fig. 9) was calculated, suggesting that Au(111) surface assistance without a Pd adatom is the preferred pathway for this addition step. In addition, the reaction energy of this exothermic step was found to be 1.85 eV, reflecting the irreversibility of the C–C bond formation. Similarly, the reaction barrier for the second addition of an imidoyl radical intermediate to another isocyanobenzene was determined to be 0.49 eV on Au (111) (Fig. 4c), affording the final product (**FS**) as a radical bonded with the Au(111) surface. The higher reaction barrier of the second addition step compared to the first addition step might be ascribed to steric hindrance.

Based on the DFT calculations, a reaction mechanism for the divergent cross-coupling of aryl bromide with isocyanide on Au (111) is proposed as depicted in Fig. 4d, in which the debromination step is the rate-limiting step. This contrasts with our previous study on the Pd-catalyzed Ullmann homo-coupling of porphyrin-substituted aryl bromide (**Br$_2$-TPP**) on Au(111), where C–C bond formation rather than the debromination step, was determined to be the rate-limiting step at high temperature. That was concluded by analyzing the isothermic reaction series and determining the bond concentration as a function of reaction temperature and duration[40]. Thus, the high selectivity of the divergent cross-coupling of **Br-TPP** with **ICBP** may be tentatively explained by the difference in reaction kinetics. For the divergent cross-coupling of **Br-TPP** with excessive **ICBP**, once the rate-limiting debromination of **Br-TPP** occurs, the subsequent two addition steps to **ICBP** will proceed spontaneously to yield final product **1**, while the homo-coupling of **Br-TPP** may be inhibited by the rate-limiting C–C formation step.

**On-surface synthesis of planar dendron and dendrimers via DCR**. The divergent cross-coupling of **Br-TPP** with **ICBP** achieved on Au(111) has three distinctive features: (1) the unique pattern enables the generation of one core and two branches, (2) high cross-coupling selectivity with complete inhibition of the homo-coupling, and (3) high cross-coupling reactivity with up to 89% abundance. These features successfully meet the challenges posed by the on-surface synthesis of planar dendrons and dendrimers.

For the on-surface synthesis of planar dendrons and dendrimers, two porphyrin precursors, **Br$_2$-TPP** and **Br$_4$-TPP**, containing two and four aryl bromides substituents, respectively, were prepared and applied to the Pd-catalyzed DCR on Au(111) (Fig. 5a, e). The STM images for the mixtures of the porphyrins with **ICBP** before and after Pd deposition are presented in Supplementary Figs. 10 and 11. Under reaction conditions similar

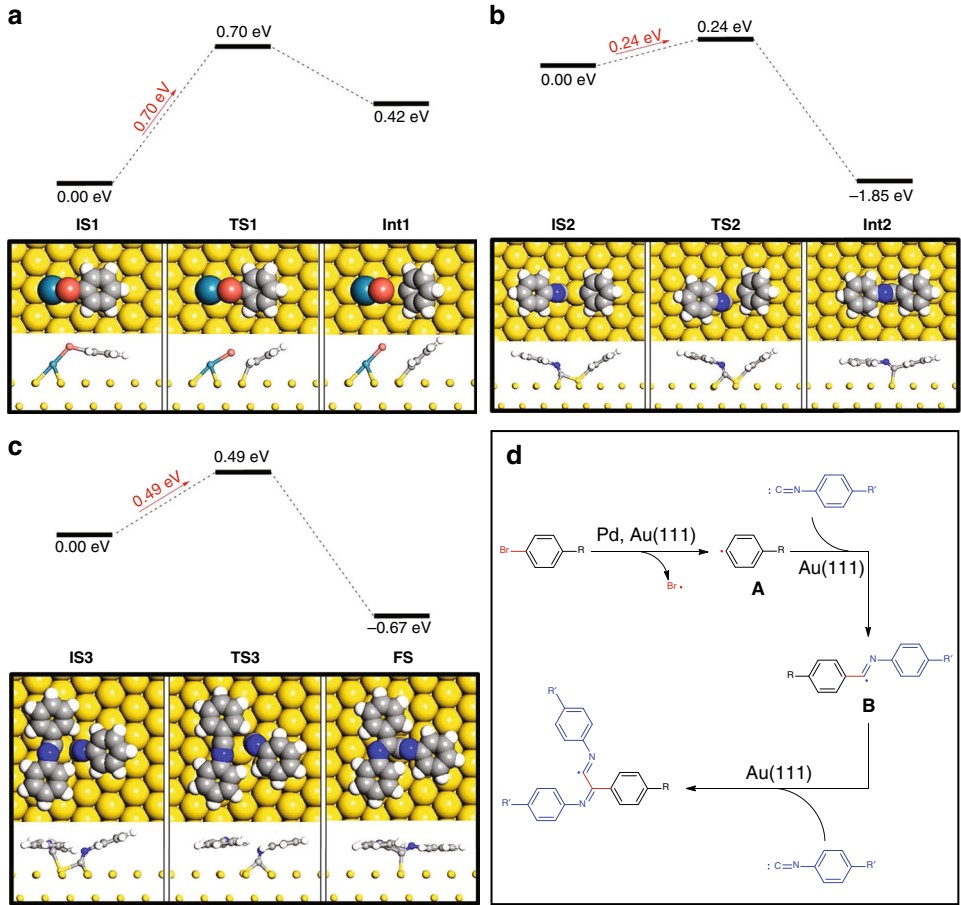

**Fig. 4** Mechanism for the divergent cross-coupling of aryl bromide with isocyanide on Au(111). **a–c** DFT-calculated energy levels and molecular structures for **a** debromination of bromobenzene, **b** the first addition of a benzene radical to isocyanobenzene, and **c** the second addition of an imidoyl radical to another isocyanobenzene. Top and side views of the initial state (**IS**), transition state (**TS**), intermediate state (**Int**), and final state (**FS**) of the reactions are shown below the energy diagrams. **d** Overview of the reaction mechanism

to those used to form **1**, the on-surface reaction of one molecule of **Br$_2$-TPP** and four molecules of **ICBP** readily proceeded to generate dendron **2** containing one porphyrin core and four branches. It is noteworthy that the selectivity was also very high, and no homo-coupling products were observed in this reaction (the large-scale STM image is presented in Supplementary Fig. 10c). As shown in the STM image, Fig. 5b, dendron **2** formed close-packed molecular islands, in which the molecules self-assembled to afford aligned herringbone-like structures similar to the self-assembled structure of **1**. From the high-resolution STM images (Fig. 5c, d), dendron **2** in the *trans*-configuration could be distinguished as *R,R*-**2** and *S,S*-**2**. Interestingly, the *cis*-dendron **2** could barely be observed in the reaction, which demonstrates the high selectivity of the reaction on the surface.

As shown in the STM image (Fig. 5f), dendrimer **3**, with a crab-like structure, was also readily achieved on Au(111) through the Pd-catalyzed divergent cross-coupling of one **Br$_4$-TPP** with eight **ICBP** molecules. Notably, no homo-coupling reaction of **Br$_4$-TPP** was observed and the reactivity of the cross-coupling reaction was very high, yielding isolated molecules of dendrimer **3** all containing one porphyrin core and eight branches (the large-scale STM image is presented in Supplementary Fig. 11c).

To achieve the on-surface synthesis of planar dendrimer with a larger size on a two-dimensional surface, we synthesized the **Br$_6$-B$_{10}$** precursor including six aryl bromides as the reactive sites, with the hope of achieving the space-crowded dendrimer **4** with 12 branches (Fig. 5g). After the successive deposition of

**Br$_6$-B$_{10}$**, **ICBP** and Pd (see Supplementary Fig. 12a) and annealing at 403 K for 1 h, dendrimer **4** with 12 symmetrical antennae was generated on Au(111) surface, as shown in Fig. 5h. Such a planar dendrimer may possess better conjugate properties because the dihedral angle between the adjacent benzene rings is significantly reduced in comparison with the dendrimers stretched in three-dimensional space. The large-scale STM image (see Supplementary Fig. 12b) suggests that the reaction selectivity was poor, and defects were prevalent in the products due to the crowdedness and strong steric hindrance in the two-dimensional confined space, which severely hampered the DCR. The results indicate that surface coverage is crucial for the on-surface synthesis of planar dendrimers. Noted that excess of hydrogen in the chamber might passive the surface-bonded radicals of the products to generate the hydrogenated species.

## Discussion

We have achieved the on-surface divergent growth reaction via cross-coupling of one molecule of aryl bromide with two molecules of isocyanide on Au(111), generating a product containing a core and two branches. Exogenous palladium has been used as a catalyst to promote the DCR with high selectivity and reactivity. Statistical analysis demonstrated that the reaction possesses good tolerance to the coverage and annealing temperature. The structure of the product incorporating a porphyrin core and two isocyanide-derived branches has been confirmed through STM measurements at single-molecule resolution and ToF-SIMS

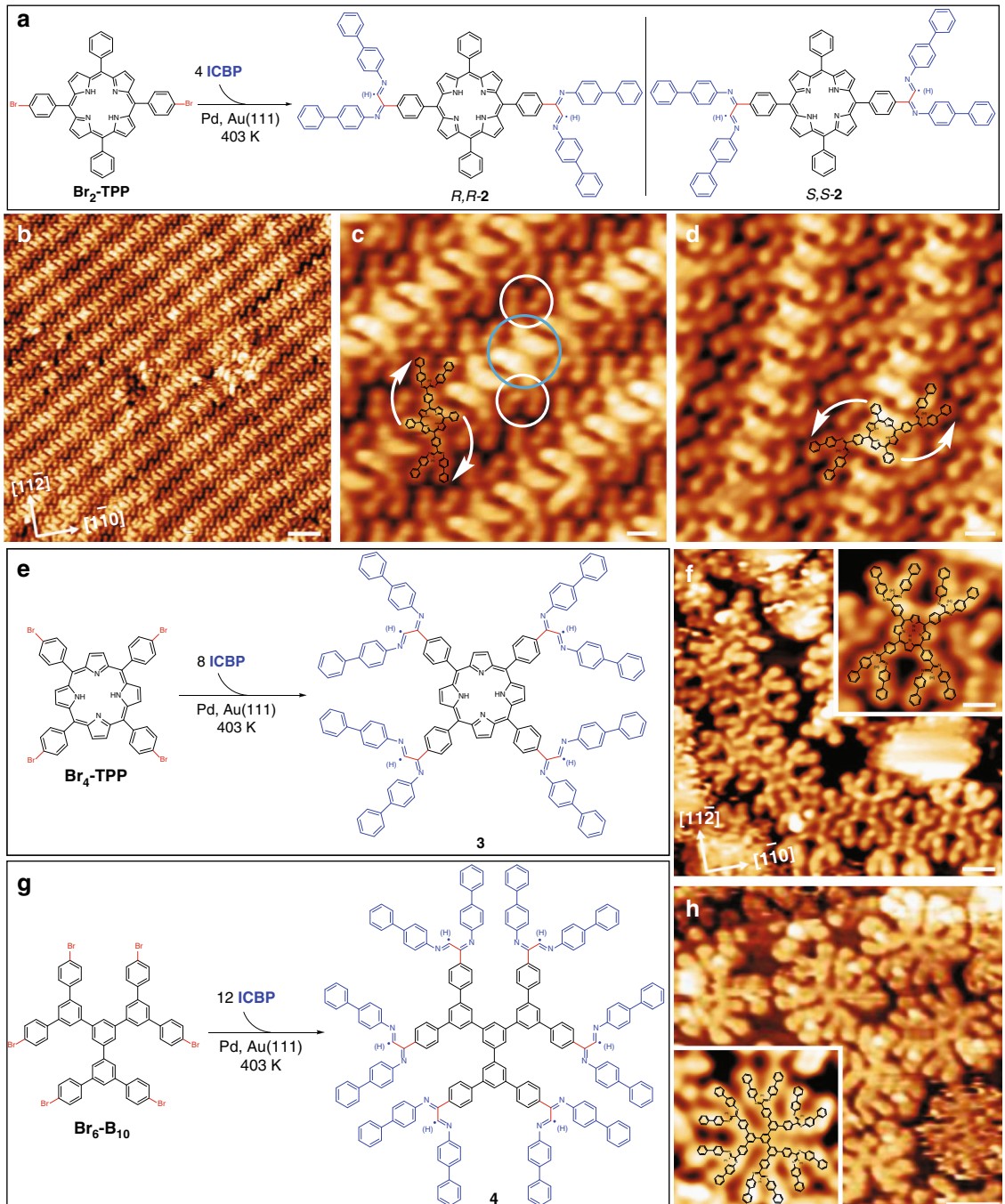

**Fig. 5** On-surface synthesis of planar dendron and dendrimers. **a** Schematic representation of DCR used to generate dendrons *R,R*-**2** and *S,S*-**2**. **b** STM image of self-assembled dendron **2** on Au(111). **c** STM image of self-assembled dendron *R,R*-**2**. **d** STM image of self-assembled dendron *S,S*-**2**. **e** Schematic representation of DCR used to generate dendrimer **3** on Au(111). **f** STM image of dendrimer **3**. The inset represents a close-up image of a single dendrimer **3**. **g** Schematic representation of DCR used to generate dendrimer **4** on Au(111). **h** STM image of dendrimer **4**. The inset represents a close-up image of a single dendrimer **4**. The dendron **2**, dendrimers **3** and **4** might be surface-bonded radicals or hydrogenated species. Scale bars: **b** 3 nm. **c**, **d** and inset of (**f**, **h**) 1 nm. **f** 2 nm. **h** 2 nm. Tunneling parameters. **b**, **c** $I = -0.11$ nA, $U = -1.80$ V. **d** $I = -0.17$ nA, $U = -1.67$ V. **f** $I = -0.52$ nA, $U = -1.52$ V. **h** $I = 0.49$ nA, $U = 1.74$ V

determination of the accurate molecular weight. The DFT calculations suggest that the reaction proceeds via palladium adatom-promoted C–Br activation and sequential addition to two isocyanides. The C–Br activation is the rate-limiting step, which guarantees the high selectivity of the cross-coupling. On the basis of this reaction, a planar dendron with four branches and dendrimers with eight and twelve branches have been synthesized on Au(111) surface in orderly self-assembled patterns. Our results demonstrate a promising protocol for the synthesis of a variety of planar dendrons and dendrimers with diverse structures and functionalities, which might offer great opportunities for future investigations on dendritic effects at the single-molecule level and the applications in functional materials and nano-devices.

## Methods

**Experimental measurement**. All experiments were conducted using commercial UHV system (base pressure $2 \times 10^{-10}$ mbar) equipped with a variable temperature scanning tunneling microscope (SPECS, Aarhus 150), a molecular evaporator, a

metal evaporator and standard facilities for sample preparation. The single-crystalline Au(111) surfaces were cleaned by cycles of argon-ion sputtering and annealing. Imaging was performed in constant current mode, with the bias voltage given with respect to the sample at ~120 K. The precursor **ICBP** was deposited through a leak valve onto the substrates. Precursors **Br-TPP**, **Br₂-TPP**, **Br₄-TPP**, and **Br₆-B₁₀** were evaporated from a quartz crucible onto metal/semimetal surfaces, and the sublimation temperatures for precursors **Br-TPP**, **Br₂-TPP**, **Br₄-TPP**, and **Br₆-B₁₀** were 165, 175, 185, and 230 °C, respectively. Palladium atoms were dosed by electron beam evaporation from a Pd rod. ToF-SIMS experiments were performed using a ToF-SIMS V spectrometer (IONTOF GmbH, Münster, Germany). A pulsed 30 keV Bi₃⁺ ion beam was used as the primary ion beam for all measurements. The analysis area was 500 × 500 μm. Target current was 1 pA. All data were obtained and analyzed using the IONTOF instrument software. Positive mass spectra were calibrated using C⁺, CH⁺, CH₂⁺, and C₂⁺ peaks.

**General procedures for the preparation of sample**. An excess of **ICBP** precursors (~0.8–1.0 ML) were deposited onto a pristine Au(111) surface held at approximately 250 K. Then, a submonolayer of aryl bromide precursor (**Br-TPP**, **Br₂-TPP**, **Br₄-TPP**, or **Br₆-B₁₀**) was deposited onto the Au(111) surface at approximately 250 K. Next, a submonolayer (~0.1 ML) of Pd atoms was dosed onto the sample containing two precursors at approximately 250 K. Then, the sample was annealed to indicated temperature for indicated time and cooled to 120 K for STM analysis.

**Theoretical calculation**. DFT calculations were performed with the Vienna Ab-initio Simulation Package (VASP)[43, 44], using the projector-augmented wave method[45, 46]. We used the generalized gradient approximation (GGA) with Perdew–Burke–Ernzerhof (PBE) formalism to treat exchange-correlation interaction[47], and van der Waals (vdW) interactions were considered by using the DFT-D3 developed by Grimme[48]. The initial debromination was performed on Au(111) surface (5 × 5), which is different from (4 × 5) in our former report[36]. The two subsequent addition steps were performed on Au(111) surface (5 × 5) and (7 × 7), respectively. Transition-state calculations according to the nudged elastic band and the climbing-image nudged elastic band was applied to locate the transition state[49]. STM simulation was performed using the Tersoff–Hamann approximation[50].

## Data availability

The authors declare that the main data supporting the findings of this study are available within the paper and its Supplementary Information files. Extra data are available from the corresponding authors upon reasonable request.

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

## Acknowledgements

The authors thank Prof. Xiao-Hui Qiu in National Center for Nanoscience and Technology for his insightful suggestions to this work. This work was supported by the National Natural Science Foundation of China (Project Nos. 21421004, 21561162003, 21672059, and 91845110), Shanghai Municipal Science and Technology Major Project (Grant No. 2018SHZDZX03), the Programme of Introducing Talents of Discipline to Universities (B16017), Program of the Shanghai Committee of Sci. & Tech. (Project No. 18520760700), and the Program for Eastern Scholar Distinguished Professor. The calculations were performed on TianHe-1 (A) at National Supercomputer Center in Tianjin.

## Author contributions

D.-Y.L., S.-W.L., and Y.-L.X. performed the STM experiments. D.-Y.L. and S.-W.L. conducted theoretical computations. A.W. synthesized the molecules. X.H. and Y.-T.L. performed the ToF-SIMS experiments. D.-Y.L., S.-W.L., and P.-N.L. analyzed the data. P.-N.L. planned and supervised the project and all work. All authors discussed the results and helped in writing the manuscript.
