## [Peer Review File · Nature Communications]

Reviewers' Comments:

Reviewer #1:

Remarks to the Author:

This manuscript describes the purported synthesis of planar dendrimers via a divergent cross-coupling procedure. In general, the manuscript has several problems from several points of view. (1) These are not dendrimers, Vögtle et al. described "cascade" compounds and only went 2 ½ levels (generations) BUT Tomalia circumvented this early work by defining (in the DOW patents) a dendrimer as a product with 3 or more levels of branching. Thus, Tomalia's (1985, 1 to 2 branching) and Newkome's (1985, 1 to 3 branching) examples were the first dendrimers. In the current paper, the authors achieved only 1 to 2 branching up to level 1 in a pure manner and the attempted 2nd generation example (4) was not pure, probably because of their G1 core was incapable of being totally flat on the gold surface. Their Figure 1a is, in fact, a G2 branched specie but 1, 2, and 3 are only G1 and 4 is G2. So these are simply branched materials analogous to Vögtle.

The authors also could simplify the manuscript by reduction of the use of meaningless words e.g. ...to steric hindrance. (delete effects)

Based on the DTF calculations, (delete results)

The divergent cross-coupling (delete reaction; a ~dozen times page 9-11 alone)

...self-assembled dendrimer or dendrimer 2 (9 times in the figure 5 caption, page 11) the word dendrimer is not needed, since there is a number to the structure of concern and as noted above these are not dendrimers.

Reviewer #2:

Remarks to the Author:

Dear Editor

The work of Liu and coworkers report on palladium-catalyzed cross-coupling between isocyanobiphenyls and arylbromides on an Au(111) surface, yielding first generation dendrimers. The coupling reaction proceeds by exposing a surface assembly of the arylbromide precursors to the isocyanobiphenyl and subsequently depositing Pd. The reaction appears to proceed with high yield at a temperature of 130 °C, with claimed near perfect yield and selectivity.

The study is of interest to the Nature Communications audience, once evidence for the extraordinary claims is provided; e.g. detailed methodological descriptions, statistics, additional STM data and complete assignment of MS data. Unlike similar studies by the authors (Chem. Eur. J., 20, 4111, 2014) the current form of the manuscript does not include statistical analysis nor systematic exploration of reaction conditions.

One extraordinary claim regards the absence of aryl-aryl Pd homocoupled reactions. According to the mechanism in Figure 4a, Pd-catalyzed homocoupling of two or more porphyrins should be enthalpically favored. Moreover, previous work places the onset of Pd-catalyzed homocoupling on surfaces at around 100 °C (Chem. Commun., 52, 13225, 2016; Chem. Eur. J., 20, 4111, 2014). Thus, it is expected that homocoupled products be present near Pd islands and in the MS data. Absence of the expected homocoupled product should be thoroughly justified, i.e. by statistical analysis of the coverage-dependent conditions employed. The employed kinetic control over the reaction should be discussed.

Another extraordinary claim regards the synthesis of several multiradical species at the surface. Product 4 is an extraordinary hexaradical. Excess of hydrogen in the chamber might 'passivate' the

radical products, yet at the same time compete with the second 'divergent' addition of isocyanobiphenyl. However, no 'partially-branched' dendrons are reported, and the divergent reaction is claimed to occur with 'high yield'. Figure 4 implies that the hexaradical is stabilized by the surface, in which case an atomistic epitaxial absorption model and corresponding STM simulation is required to conform to the high scholarly standard of Nature Communications.

Finally, the authors should provide overview, large area data of the porphyrin/precursor assemblies before and after palladium deposition. This information has been provided by the authors in similar studies, where the cross-coupling reaction occurred at 150 °C (ref. 36). Thus, it is clear that overview STM data will help understand and reproduce the high-yield, high-selectivity, kinetic/coverage-dependent synthesis of multiradical species under competing homocoupling conditions, especially when systematic exploration of temperature and coverage conditions is missing.

Given the manuscript's current data, it can be argued that metal-organic complexes of palladium are present or that palladium is not necessary for the reaction to occur, that is, 'excess' of (potentially pre-dimerized) isocyanobiphenyl readily reacts with arylbromides.

Minor details

- o How was the temperature read in this study compared to the previous (Chem. Eur. J., 20, 4111, 2014) Pd-catalyzed study? Sample temperature can greatly vary between setups.
- o A structure in Fig. S1 is incomplete.
- o It is advice to replace 'space compatibility' (and similar non-scientific terminology) with proper scientific language, e.g. 'surface coverage'.
- o Figure 1a depicts four branches per divergent point, while only two branches are explored in the present study.
- o Language issues can be further addressed, e.g. 'The product highly orderly self-assembled' may read 'The resulting highly ordered self-assembly' and similar.
- o The mechanism in Fig. 4a appears identical to Fig. 3a in reference 36. Why is it now different? Does this imply an intrinsic conformational statistical error of at least 0.13 eV? If so, the statistical error should be shown in Fig. 4a.

Reply to reviewers' comments

We sincerely thank the referees for their careful reading and suggestions for improving our paper. We have revised the manuscript on the basis of these comments. The detailed responses are attached below.

Reviewer #1:

1. This manuscript describes the supported synthesis of planar dendrimers via a divergent cross-coupling procedure. In general, the manuscript has several problems from several points of view. (1) These are not dendrimers, Vögtle et al. described "cascade" compounds and only went 2 ½ levels (generations) BUT Tomalia circumvented this early work by defining (in the DOW patents) a dendrimer as a product with 3 or more levels of branching. Thus, Tomalia's (1985, 1 to 2 branching) and Newkome's (1985, 1 to 3 branching) examples were the first dendrimers. In the current paper, the authors achieved only 1 to 2 branching up to level 1 in a pure manner and the attempted 2nd generation example (4) was not pure, probably because of their G1 core was incapable of being totally flat on the gold surface. Their Figure 1a is, in fact, a G2 branched specie but 1, 2, and 3 are only G1 and 4 is G2. So these are simply branched materials analogous to Vögtle.

Reply: We appreciate the insightful comments to help us improve this manuscript! Accordingly, the products **1** and **2** are two single dendrons, product **3** is the 1st generation dendrimer and product **4** is the 2nd dendrimer (ref. 4: *Chem. Rev.* **109**, 6047–6076 (2009); ref. 10: *Chem. Rev.* **110**, 1857–1959 (2010); ref.12: *Chem. Soc. Rev.* **47**, 514-532 (2018)). The 2nd generation dendrimer **4** obtained is not pure due to the crowdedness and strong steric hindrance on the two-dimensional confined space, which severely hamper the divergent cross-coupling reaction. In the manuscript, relevant descriptions of products **1-4** have been revised accordingly. In page 13, line 4, “Notably, the reaction selectivity was poor and defects were prevalent in the products due to the strong steric hindrance effect in the two-dimensional confined space.” has been changed to “The large scale STM image (see Supplementary Fig. 9b) suggests that the reaction selectivity was poor and defects were prevalent in the products due to the crowdedness and strong steric hindrance in the two-dimensional confined space, which severely hampered DCR.”

2. The authors also could simplify the manuscript by reduction of the use of

meaningless words e.g.

...to steric hindrance. (delete effects)

Based on the DTF calculations, (delete results)

The divergent cross-coupling (delete reaction; a ~dozen times page 9-11 alone)

...self-assembled dendrimer or dendrimer 2 (9 times in the figure 5 caption, page 11)

the word dendrimer is not needed, since there is a number to the structure of concern and as noted above these are not dendrimers.

Reply: Thanks a lot for the comments. The manuscript has been revised accordingly and highlighted.

Reviewer #2:

1. The work of Liu and coworkers report on palladium-catalyzed cross-coupling between isocyanobiphenyls and arylbromides on an Au(111) surface, yielding first generation dendrimers. The coupling reaction proceeds by exposing a surface assembly of the arylbromide precursors to the isocyanobiphenyl and subsequently depositing Pd. The reaction appears to proceed with high yield at a temperature of 130 °C, with claimed near perfect yield and selectivity.

The study is of interest to the Nature Communications audience, once evidence for the extraordinary claims is provided; e.g. detailed methodological descriptions, statistics, additional STM data and complete assignment of MS data. Unlike similar studies by the authors (Chem. Eur. J., 20, 4111, 2014) the current form of the manuscript does not include statistical analysis nor systematic exploration of reaction conditions.

Reply: We appreciate the referee's positive and helpful comments. In Page 14, paragraph 2, "**General procedures for the preparation of sample.** An excess of **ICBP** precursors (~0.8-1.0 ML) were deposited onto a pristine Au(111) surface held at approximately 250 K. Then, a submonolayer of aryl bromide precursor (**Br-TPP**, **Br₂-TPP**, **Br₄-TPP** or **Br₆-B₁₀**) was deposited onto the Au(111) surface at approximately 250 K. Next, a submonolayer (~0.1 ML) of Pd atoms was dosed onto the sample containing two precursors at approximately 250 K. Then, the sample was annealed to indicated temperature for indicated time and cooled to 120 K for STM analysis." has been added.

For the comments of statistical analysis for systematic exploration of reaction conditions, Figures 2f-h have been added. In page 6, last paragraph, "To further demonstrate the selectivity of the DCR, we performed statistical analysis for the systematic exploration of the coverage of **Br-TPP**, annealing temperature and time.

As shown in Figure 2f, the abundance of product **1** slightly decreased when the coverage of **Br-TTP** increased from 0.2 ML to 0.6 ML, but dramatically decreased once the coverage increased to 0.8 ML, due to the limited two-dimensional space. Figure 2g shows that an annealing temperature of 378 K led to a slightly lower abundance of product **1**, while lowering the annealing temperature to 353 K dramatically decreased the abundance of product **1**. In contrast, enhancement of the annealing temperature demonstrated no obvious influence on the reaction. The results indicated that the reaction possesses good tolerance to the coverage of **Br-TTP** and annealing temperature. The investigation of annealing time at 403 K showed that the reaction was rather fast, and the abundance of the product was 77% after 10 min and increased to 82% after 15 min (Fig. 2h). For the Pd-catalyzed Ullmann homo-coupling of the porphyrin-substituted aryl bromide on Au(111), the homo-coupling reaction is more sluggish because the C–C bond formation, rather than the debromination step, is the rate-limiting step.⁴⁰ However, the results of DCR suggest that the coupling step is fast, which guarantees the high selectivity of the reaction with the inhibition of the homo-coupling reactions.” has been added.

For the complete assignment of MS data, the Figure 3 has been revised and Supplementary Figure 5 has been added. In page 7, last paragraph, “It is noteworthy that the lower abundance peaks at $m/z = 794.33$ and 615.25 amu correspond accurately to the quasi-molecular ion peaks (see Supplementary Fig. 5) for the partially branched product of the cross-coupling of one **Br-TTP** with one **ICBP** ($C_{57}H_{40}N_5^+$, $[M_{CR}+H]^+$, calculated value is 794.33) and the debrominated product of **Br-TTP** ($C_{44}H_{31}N_4^+$, $[M_{TTP}+H]^+$, calculated value is 615.25), respectively. In contrast, molecular ion peaks of the homo-coupling products of **Br-TTP** or **ICBP** were not observed.” has been added.

2. One extraordinary claim regards the absence of aryl-aryl Pd homocoupled reactions. According to the mechanism in Figure 4a, Pd-catalyzed homocoupling of two or more porphyrins should be enthalpically favored. Moreover, previous work places the onset of Pd-catalyzed homocoupling on surfaces at around 100 °C (Chem. Commun., 52, 13225, 2016; Chem. Eur. J., 20, 4111, 2014). Thus, it is expected that homocoupled products be present near Pd islands and in the MS data. Absence of the expected homocoupled product should be thoroughly justified, i.e. by statistical analysis of the coverage-dependent conditions employed. The employed kinetic control over the reaction should be discussed.

Reply: Thanks a lot for the insightful comments. In page 6, the 1st paragraph, “The results demonstrate that the alternative arrangement of **ICBP** and **Br-TTP** was mostly retained after the Pd deposition and Pd clusters scattered in the molecules. Such miscibility of the reactants is crucial for the high selectivity of the DCR.” has been added. In page 7, the 1st paragraph, “For the Pd-catalyzed Ullmann homo-coupling of the porphyrin-substituted aryl bromide on Au(111), the homo-coupling reaction is more sluggish because the C–C bond formation, rather than the debromination step, is the rate-limiting step.⁴⁰ However, the results of DCR suggest that the coupling step is fast, which guarantees the high selectivity of the reaction with the inhibition of the homo-coupling reactions.” has been added.

Moreover, in page 9, the last paragraph, “Thus, the high selectivity of the divergent cross-coupling of **Br-TTP** with **ICBP** may be tentatively explained by the difference in reaction kinetics. For the divergent cross-coupling of **Br-TTP** with excessive **ICBP**, once the rate-limiting debromination of **Br-TTP** occurs, the subsequent two addition steps to **ICBP** will proceed spontaneously to yield final product **1**, while the homo-coupling of **Br-TTP** may be inhibited by the rate-limiting C–C formation step.” is also the explanation for the high selectivity.

3. Another extraordinary claim regards the synthesis of several multiradical species at the surface. Product 4 is an extraordinary hexaradical. Excess of hydrogen in the chamber might ‘passivate’ the radical products, yet at the same time compete with the second ‘divergent’ addition of isocyanobiphenyl. However, no ‘partially-branched’ dendrons are reported, and the divergent reaction is claimed to occur with ‘high yield’. Figure 4 implies that the hexaradical is stabilized by the surface, in which case an atomistic epitaxial absorption model and corresponding STM simulation is required to conform to the high scholarly standard of Nature Communications.

Reply: Thanks a lot for the insightful comments. In the statistical analysis of the STM data, a small amount of partially branched dendrons were found as the product of CR (see Fig. 2f-h), as well as in MS data. In the manuscript, Figures 2f-h have been added and the relevant discussion has been added in page 6, last paragraph. In Figure 3, the MS data has been revised and the existence of the partially branched dendron has been assigned $[M_{CR+H}]^+$. In page 7, last paragraph, “It is noteworthy that the lower abundance peaks at $m/z = 794.33$ and 615.25 amu correspond accurately to the quasi-molecular ion peaks (see Supplementary Fig. 5) for the partially branched product of the cross-coupling of one **Br-TTP** with one **ICBP** ($C_{57}H_{40}N_5^+$, $[M_{CR+H}]^+$,

calculated value is 794.33) and the debrominated product of **Br-TTP** ($C_{44}H_{31}N_4^+$, $[M_{TTP}+H]^+$, calculated value is 615.25), respectively.” has been added.

Because of the large size of dendrimer **4**, the adsorption site of **4** can not be determined accurately on surface and the DFT calculation for atomistic epitaxial absorption model is very difficult due to the limitation of calculation resource. Instead, we performed the DFT calculation for the relevant hydrogenated dendrimer **4** in vacuum and gave the model and simulated STM image in Supplementary Figure 10.

4. Finally, the authors should provide overview, large area data of the porphyrin/precursor assemblies before and after palladium deposition. This information has been provided by the authors in similar studies, where the cross-coupling reaction occurred at 150 °C (ref. 36). Thus, it is clear that overview STM data will help understand and reproduce the high-yield, high-selectivity, kinetic/coverage-dependent synthesis of multiradical species under competing homocoupling conditions, especially when systematic exploration of temperature and coverage conditions is missing.

Reply: Thanks a lot for the insightful comments. The overview, large area datas of the porphyrin/precursor assemblies before and after palladium deposition have been added to Supplementary Figures 3, 4, and 7-9. Moreover, the systematic exploration of temperature and coverage conditions has been performed and the revisions are listed in the former reply. In page 5, last paragraph, “A submonolayer of **Br-TTP** molecules and an excess of **ICBP** molecules were successively deposited onto a pristine Au(111) surface” has been changed to “A submonolayer of **Br-TTP** molecules (see Supplementary Fig. 3a) and an excess of **ICBP** molecules (see Supplementary Fig. 4a) were successively deposited onto a pristine Au(111) surface (see Supplementary Fig. 4b)”. In page 6, the 1st paragraph, “Pd atoms were subsequently dosed onto the Au(111) surface” has been changed to “Pd atoms were subsequently dosed onto the Au(111) surface (see Supplementary Fig. 4c).” In page 12, the 2nd paragraph, “The STM images for the mixtures of them with **ICBP** before and after Pd deposition were presented in Supplementary Figures 7 and 8.” has been added. In page 12, the last paragraph, “After the successive deposition of **Br₆-B₁₀**, **ICBP** and Pd (see Supplementary Fig. 9a) and annealing at 403 K for 1 h,” has been added.

5. Given the manuscript's current data, it can be argued that metal-organic complexes

of palladium are present or that palladium is not necessary for the reaction to occur, that is, 'excess' of (potentially pre-dimerized) isocyanobiphenyl readily reacts with arylbromides.

Reply: Thanks a lot for the comments. The palladium is necessary and no cross-coupling reaction occurs under 403 K without palladium. In the supporting information, the reaction result without palladium has been added as Supplementary Figure 2. In page 4, last paragraph, “The experimental results showed that no cross-coupling reaction occurred between the porphyrin derived aryl bromide and the isocyanide on Au(111) after annealing to 403 K.” has been changed to “The experimental results showed that no cross-coupling reaction occurred between the porphyrin derived aryl bromide and the isocyanide on Au(111) after annealing to 403 K (see Supplementary Fig. 2).”

6. *How was the temperature read in this study compared to the previous (Chem. Eur. J., 20, 4111, 2014) Pd-catalyzed study? Sample temperature can greatly vary between setups.*

Reply: Thanks a lot for the comment. The annealing temperature of two setups is similarly measured by the thermocouples, so the temperature is comparable although variation should surely exist.

7. *A structure in Fig. S1 is incomplete.*

Reply: Thanks a lot for the reminder. In Supplementary Figure 1, incomplete structure of precursor **Br₄-TPP** has been revised accordingly.

8. *It is advice to replace 'space compatibility' (and similar non-scientific terminology) with proper scientific language, e.g. 'surface coverage'.*

Reply: Thanks a lot for the comment. In page 13, the 1st paragraph, “space compatibility” has been changed to “compactibility of the space”.

9. *Figure 1a depicts four branches per divergent point, while only two branches are explored in the present study.*

Reply: Thanks a lot for the comment. Figure 1a is just a schematic diagram of the structure of general dendrimers, not only the schematic diagram for the dendrimers in the present study.

10. *Language issues can be further addressed, e.g. 'The product highly orderly self-assembled' may read 'The resulting highly ordered self-assembly' and similar.*

Reply: Thanks a lot for the helpful suggestion. In page 6, the second paragraph, “The product highly orderly self-assembled” has been revised to “The resulting highly ordered self-assembly”.

11. *The mechanism in Fig. 4a appears identical to Fig. 3a in reference 36. Why is it now different? Does this imply an intrinsic conformational statistical error of at least 0.13 eV? If so, the statistical error should be shown in Fig. 4a.*

Reply: Thanks a lot for the insightful comment. For the debromination step, the unit cell on Au(111) surface in current work is (5×5) and the unit cell in ref. 36 is (4×5), which may cause the energy difference. We think the current result is the more optimized one. In page 14, in the part of “Theoretical calculation”, “The initial debromination was performed on Au(111) surface (5×5)” has been changed to “The initial debromination was performed on Au(111) surface (5×5), which is different from (4×5) in our former report.³⁶”

Reviewers' Comments:

Reviewer #1:

Remarks to the Author:

I am sorry but page 2 line 30 is not correct "Ever since the first dendrimer was reported by Vögtle et al. in 1978,...", since dendrimers were actually started in 1985. I suggest: Ever since the first cascade molecules, nee dendrimers, were reported by Vögtle et al. in 1978,... [This is actually the way it was...]

Page 2 line 41, 42 ...produce highly ordered, self-assembled structures. In addition, the in situ synthesis of dendrimers...

Starting with page 4 ...cross-coupling reaction... ("reaction" is not necessary) check throughout the manuscript

Starting with page 6, line 121 ...generate product 1 ... thereafter the word "product" is unnecessary (used 12+ times...)

Page 11 ...a single dendrimer molecule 4. Again "Molecule" adds nothing ...

Reviewer #2:

Remarks to the Author:

Dear Editor

The work of Li and coworkers has significantly improved and might now be reproducible, as the reaction conditions have been amended. It is strongly recommended that the authors carefully address the previous important extraordinary claims, which could potentially be misleading.

To address their first extraordinary claim; the high reaction selectivity without competing homocoupling reactions; the authors have included statistical analysis, large-scale data and partially assigned their MS data. Thanks to Suppl. Figure 4d, the readership has now the possibility to conclude on the veracity of the claim. It is now plausible that the homocoupling reaction is kinetically inhibited because of the unique phase mixing (bicomponent self-assembly) of the reactants (Suppl. Figure 4c). Figure 2f corroborates that the BCR does not proceed in absence of phase mixing, i.e., at high Br-TPP coverages. At such high coverage, homocoupling is expected and STM data at this coverage must be provided. In this regard, the authors are encouraged to mention in the caption of Figure 3 the coverage conditions employed for acquiring the mass spectrum and assign the potential homocoupling product peaks between 1100 and 1300 m/z in the mass spectrum.

To summarize, based on this work and on Chem. Eur. J., 20, 4111, 2014, this reviewer can conclude from the data available that; on Au(111) and for brominated TPPs:

1. Homocoupling reaction proceeds at ~411 K (Chem. Eur. J., 20, 4111, 2014).
2. Exerting kinetic control via diluted reactant conditions inhibits homocoupling products, thereby kinetically selecting the DCR product.

However, an extraordinary claim by the authors remains, also potentially contradicting Chem. Eur. J., 20, 4111, 2014. Figure 2f claims that there is no reaction (implying no homocoupling reaction) at high coverage (0.8 ML) and 403 K annealing. Thus, the authors effectively claim that:

3. Homocoupling reaction proceeds at ~411 K (Chem. Eur. J., 20, 4111, 2014) but not at 403 K (This work).

A <7° thermodynamic selectivity is obviously extraordinary. The authors further clarify in their response that their temperature measurement should be identical to Chem. Eur. J., 20, 4111, 2014.

The authors should very carefully clarify this, citing their Chem. Eur. (e.g. stating 'unlike Chem. Eur...') and adopting appropriate terminology to justify what they believe are their reaction conditions.

The authors did not address their claim of radical formation at interfaces. Are they synthesizing radical/multiradical dendrimers? At the moment this is shown in their schematics and implied by DFT. Moreover, it appears that the DCR model is evidenced as a radical species from the MS, with a mass of 971.39 (the hydrogen passivated structure is 972.39). Accordingly, the mass of the non-radical CR species is 793.32+H. If the dendrimers are believed to be multiradical, this needs to be thoroughly discussed, and if stabilization by the surface is claimed (which appears to be the case by DFT), an epitaxial/adsorption toy model must be shown for the molecule. This can be constructed by superposing their simulated STM data (Figure 2b) on top of an Au(111) grid. This is pot. a requirement, as claiming the synthesis of an hexaradical such as product 4, appears truly extraordinary. Otherwise the authors should adapt the manuscript's chemical structures to show the plausible hydrogenation of their products.

Minor details

- o Why does the mass spectrum show >50% unreacted species yet this situation is not observed in STM?
- o "compactibility of the space " is neither defined nor a known term.

Reply to reviewers' comments

We sincerely thank the referees for their careful reading and suggestions for improving our paper. We have revised the manuscript on the basis of these comments. The detailed responses are attached below.

Reviewers' comments:

Reviewer #1 :

I am sorry but page 2 line 30 is not correct "Ever since the first dendrimer was reported by Vögtle et al. in 1978,"; since dendrimers were actually started in 1985. I suggest: Ever since the first cascade molecules, nee dendrimers, were reported by Vögtle et al. in 1978,; [This is actually the way it was;]

Page2 line 41, 42; produce highly ordered, self-assembled structures. In addition, the in situ synthesis of dendrimers.....;

Starting with page 4; cross-coupling reaction; ("reaction" is not necessary) check throughout the manuscript

Starting with page 6, line 121; generate product 1; thereafter the word "product" is unnecessary (used 12+ times;)

Page 11; a single dendrimer molecule 4. Again "Molecule" adds nothing;

Reply: Thanks a lot for the comments to help us improve this manuscript. Accordingly, we have revised the manuscript as highlighted throughout the manuscript.

Reviewer #2:

1. The work of Li and coworkers has significantly improved and might now be reproducible, as the reaction conditions have been amended. It is strongly recommended that the authors carefully address the previous important extraordinary claims, which could potentially be misleading.

To address their first extraordinary claim; the high reaction selectivity without competing homocoupling reactions; the authors have included statistical analysis, large-scale data and partially assigned their MS data. Thanks to Suppl. Figure 4d, the readership has now the possibility to conclude on the veracity of the claim. It is now plausible that the homocoupling reaction is kinetically inhibited because of the unique phase mixing (bicomponent self-assembly) of the reactants (Suppl. Figure 4c). Figure 2f corroborates that the BCR does not proceed in absence of phase mixing, i.e., at high Br-TPP coverages. At such high coverage, homocoupling is expected and STM

data at this coverage must be provided. In this regard, the authors are encouraged to mention in the caption of Figure 3 the coverage conditions employed for acquiring the mass spectrum and assign the potential homocoupling product peaks between 1100 and 1300 m/z in the mass spectrum.

Reply: We appreciate and agree the referee's insightful comments. At high coverage of **Br-TPP**, the STM image has been added to Supplementary Figures 6. In page 7, paragraph 2, "but dramatically decreased once the coverage increased to 0.8 ML, due to the limited two-dimensional space." has been changed to "Once the coverage increased to 0.8 ML, the abundance of **1** dramatically decreased and most molecules were **TPP** species without coupling, accompanied by a very small amount of homo-coupling product **TPP-TPP** (see Supplementary Fig. 6). This result might be attributed to the high surface coverage that led to the compactly packed porphyrins and hampered both the cross- and homo-coupling reactions."

In the caption of Fig. 3, line 2, "(coverage of **Br-TPP**: ~0.5 ML)" has been added. In page 8, paragraph 2, "In contrast, molecular ion peaks of the homo-coupling products of **Br-TPP** or **ICBP** were not observed." has been changed to "In contrast, molecular ion peak of the homo-coupling product of **Br-TPP** ($C_{88}H_{59}N_8^+$ [$M_{\text{TPP-TPP}}+H$] $^+$, the calculated value is 1227.49) was not observed (see Supplementary Fig. 8)." The zoom-in mass spectrum between 1100 and 1300 m/z has also been added to Supplementary Figures 8.

2. To summarize, based on this work and on Chem. Eur. J., 20, 4111, 2014, this reviewer can conclude from the data available that; on Au(111) and for brominated TPPs:

- 1. Homocoupling reaction proceeds at ~411 K (Chem. Eur. J., 20, 4111, 2014).*
- 2. Exerting kinetic control via diluted reactant conditions inhibits homocoupling products, thereby kinetically selecting the DCR product.*

However, an extraordinary claim by the authors remains, also potentially contradicting Chem. Eur. J., 20, 4111, 2014. Figure 2f claims that there is no reaction (implying no homocoupling reaction) at high coverage (0.8 ML) and 403 K annealing. Thus, the authors effectively claim that:

- 3. Homocoupling reaction proceeds at ~411 K (Chem. Eur. J., 20, 4111, 2014) but not at 403 K (This work).*

Additinally; thermodynamic selectivity is obviously extraordinary. The authors further clarify in their response that their temperature measurement should be

identical to Chem. Eur. J., 20, 4111, 2014. The authors should very carefully clarify this, citing their Chem. Eur. (e.g. stating 'unlike Chem. Eur..') and adopting appropriate terminology to justify what they believe are their reaction conditions.

Reply: We greatly appreciate the insightful comments! In page 6, paragraph 1, “These results demonstrate that exerting kinetic control via diluted reactant conditions inhibits homo-coupling products, thereby kinetically selecting the DCR product.” has been added. In page 7, paragraph 2, “For the Pd-catalyzed Ullmann homo-coupling of the porphyrin-substituted aryl bromide on Au(111), the homo-coupling reaction is more sluggish because the C–C bond formation, rather than the debromination step, is the rate-limiting step.⁴⁰ However, the results of DCR suggest that the coupling step is fast,” has been changed to “The control experiment for Pd-catalyzed homo-coupling of **Br-TPP** without **ICBP** at 403 K was carried out and approximate 10% abundance of homo-coupling product **TPP-TPP** was obtained after 15 min, similar with our former study performed in Omicron system,⁴⁰ although the system error in temperature measurement may exist between two setups. The results suggested that the homo-coupling reaction of **Br-TPP** is sluggish, because the C–C bond formation, rather than the debromination step, is the rate-limiting step.⁴⁰ In contrast, the coupling step in DCR is fast.”.

The work in *Chem. Eur. J.*, **2014**, *20*, 4111 was performed in UHV VT STM system of Omicron in Nian Lin’s group in Hong Kong and the current work was performed in UHV VT STM system of Specs in my group in Shanghai. To further clarify the system error in temperature measurement of two setups, we have repeated the experiments of Pd-catalyzed homo-coupling of **Br₂-TPP** at 403 K and found that the homo-coupling proceeded smoothly. So, there is no thermodynamic selectivity and the system error in temperature measurement exists between two setups.

3. The authors did not address their claim of radical formation at interfaces. Are they synthesizing radical/multiradical dendrimers? At the moment this is shown in their schematics and implied by DFT. Moreover, it appears that the DCR model is evidenced as a radical species from the MS, with a mass of 971.39 (the hydrogen passivated structure is 972.39). Accordingly, the mass of the non-radical CR species is 793.32+H. If the dendrimers are believed to be multiradical, this needs to be thoroughly discussed, and if stabilization by the surface is claimed (which appears to be the case by DFT), an epitaxial/adsorption toy model must be shown for the molecule. This can be constructed by superposing their simulated STM data (Figure

2b) on top of an Au(111) grid. This is pot. a requirement, as claiming the synthesis of an hexaradical such as product 4, appears truly extraordinary. Otherwise the authors should adapt the manuscript's chemical structures to show the plausible hydrogenation of their products.

Reply: Thanks a lot for the insightful suggestions. In page 6, paragraph 2, “Through the comparison of experimental STM images with the simulated images based on density functional theory (DFT) calculations (Fig. 2b, inset),” has been revised to “Through the comparison of experimental STM images with the simulated images based on density functional theory (DFT) calculations (the adsorption models and simulated image, see Supplementary Fig. 5).” In page 14, paragraph 1, “Noted that excess of hydrogen in the chamber might passive the surface-bonded radicals of the products to generate the hydrogenated species.” has been added. Additionally, the chemical structures in the manuscript have also been revised to show the two plausible product: radicals and their hydrogenation.

4. Why does the mass spectrum show >50% unreacted species yet this situation is not observed in STM?

Reply: Thanks a lot for the comment. It is a general rule that the relative abundance of ions is relevant with the different concentration of ions, but irrelevant with the ratio of organic compounds because the ionization of different organic compounds are affected by many factors such as their polarity, molecular weight, elemental species composition, structures etc. In the MS analysis of organic compounds in solution, sometimes we find that the 1% impurity have even higher peak than the main compound. So, the abundance of mass spectrometry cannot be used to quantitatively determine the proportion of different species.

5. compactibility of the space is neither defined nor a known term.

Reply: Thanks a lot for the comment. In page 13, at the end of the 1st paragraph, “compactibility of the space” has been changed to “surface coverage”.

Reviewers' Comments:

Reviewer #2:

Remarks to the Author:

Dear Editor

The authors have now justified claims and improved aspects related to reproducibility, including the actual temperature employed, coverage dependent conditions, homocoupling under similar-previously published-reaction conditions, and debated over non-quantifying aspects of mass spectrometry data.

Regarding one of these aspects, the authors now write "The control experiment for Pd-catalyzed homo-coupling of Br-TPP without ICBP at 403 K was carried out and approximate 10% abundance of homo-coupling product TPP-TPP was obtained after 15 min". Yet for the preparation at 0.8 coverage and 403 K in Figure 2f, the NR, DCR and CR fractions amount to ~100%. In other words, there is no ~10% homocoupling product at 0.8 coverage and 403 K, implying that there is another unknown reaction condition at play in the control experiment or that ICBP hinders homocoupling.

Because of such details concerning precision and scholarly presentation, the authors are encouraged to follow the editor's guidelines with respect to the use of standard deviations for every reaction parameter (temperature, coverage, reaction products, etc.) and availability of the original statistical data set.

Following this, I can now recommend this study for publication in Nature Communications.

Reply to reviewers' comments

We sincerely thank the referees for their careful reading and suggestions for improving our paper. We have revised the manuscript on the basis of these comments. The detailed responses are attached below.

Reviewers' comments:

Reviewer #2:

The authors have now justified claims and improved aspects related to reproducibility, including the actual temperature employed, coverage dependent conditions, homocoupling under similar (previously published) reaction conditions, and debated over non-quantifying aspects of mass spectrometry data.

Regarding one of these aspects, the authors now write "The control experiment for Pd-catalyzed homo-coupling of Br-TPP without ICBP at 403 K was carried out and approximate 10% abundance of homo-coupling product TPP-TPP was obtained after 15 min". Yet for the preparation at 0.8 coverage and 403 K in Figure 2f, the NR, DCR and CR fractions amount to ~100%. In other words, there is no ~10% homocoupling product at 0.8 coverage and 403 K, implying that there is another unknown reaction condition at play in the control experiment or that ICBP hinders homocoupling.

Because of such details concerning precision and scholarly presentation, the authors are encouraged to follow the editor's guidelines with respect to the use of standard deviations for every reaction parameter (temperature, coverage, reaction products, etc.) and availability of the original statistical data set.

Following this, I can now recommend this study for publication in Nature Communications.

Reply: Thanks a lot for the insightful comments! In page 7, paragraph 2, "The control experiment for Pd-catalyzed homo-coupling of **Br-TPP** without **ICBP** at 403 K was carried out and approximate 10% abundance of homo-coupling product TPP-TPP was obtained after 15 min, similar with our former study performed in Omicron system,⁴⁰ although the system error in temperature measurement may exist between two setups." has been revised "The control experiment for Pd-catalyzed homo-coupling of **Br-TPP** without **ICBP** at 403 K for 15 min was carried out and approximate 10% abundance of homo-coupling product **TPP-TPP** was obtained in lower coverage of **Br-TPP** (~0.2 ML), similar with our former study performed in Omicron system,⁴⁰ although the system error in temperature measurement may exist between two setups. Moreover, the higher coverage of **Br-TPP** (~0.8 ML) led to lower abundance of

TPP-TPP due to the surface coverage.”. Additionally, “To further demonstrate the selectivity of the DCR, we performed statistical analysis for the systematic exploration of the coverage of **Br-TPP**, annealing temperature and time.” has been revised to “To further demonstrate the selectivity of the DCR, we performed statistical analysis for the systematic exploration of the coverage of **Br-TPP**, annealing temperature and time, in which homo-coupling product was not counted due to the low abundance of <2% (Figure 2f-h)”.